

# Effectiveness of various bioreactors for thraustochytrid culture and production (*Aurantiochytruim limacinum* BUCHAXM 122)

Khanoksinee Sirirak[1], Sorawit Powtongsook[2,3], Sudarat Suanjit[4] and Somtawin Jaritkhuan[5]

[1] Graduate Program in Environmental Science, Faculty of Science, Burapha University, Chon Buri, Thailand
[2] Center of Excellence for Marine Biotechnology, Department of Marine Science, Faculty of Science, Chulalongkorn University, Bangkok, Thailand
[3] National Center for Genetic Engineering and Biotechnology, National Science and Technology Development Agency, Pathum Thani, Thailand
[4] Department of Microbiology, Faculty of Science, Burapha University, Chon Buri, Thailand
[5] Department of Aquatic Science, Faculty of Science, Burapha University, Chon Buri, Thailand

Corresponding author
Somtawin Jaritkhuan,
somtawin@buu.ac.th

## ABSTRACT

This study aimed to develop bioreactors for cultivation of thraustochytrid, *Aurantiochytrium limacinum* BUCHAXM 122, that are low in cost and simple to operate. Obtaining maximum biomass and fatty acid production was a prerequisite. Three bioreactor designs were used: stirred tank bioreactor (STB), bubble bioreactor (BB) and internal loop airlift bioreactor (ILAB). The bioreactors were evaluated for their influence on oxygen mass transfer coefficient ($k_La$), using various spargers, mixing speed, and aeration rates. Biomass and DHA production from STB, BB, ILAB were then compared with an incubator shaker, using batch culture experiments. Results showed that a bundle of eight super-fine pore air stones was the best type of aeration sparger for all three bioreactors. Optimal culture conditions in STB were 600 rpm agitation speed and 2 vvm aeration rate, while 2 vvm and 1.5 vvm aeration provided highest biomass productivity in BB and ILAB, respectively. Antifoam agent was needed for all reactor types in order to reduce excessive foaming. Results indicated that with optimized conditions, these bioreactors are capable of thraustochytrid cultivation with a similar efficiency as cultivation using a rotary shaker. STB had the highest $k_La$ and provided the highest biomass of 43.05 ± 0.35 g/L at 48 h. BB was simple in design, had low operating costs and was easy to build, but yielded the lowest biomass (27.50 ± 1.56 g/L). ILAB, on the other hand, had lower $k_La$ than STB, but provided highest fatty acid productivity, of 35.36 ± 2.51% TFA.

## INTRODUCTION

Thraustochytrids are unicellular, eukaryote, heterotrophic and obligate marine microorganisms. They are widely distributed and play an important role as ecological

decomposers in marine environments (*Marchan et al., 2018*). They have the ability to produce high amounts of long-chain polyunsaturated fatty acids (PUFAs), such as docosahexaenoic acid (DHA) and docosapentaenoic acid (DPA), essential fatty acids for marine animals (*Nakai & Naganuma, 2015*; *Unagul et al., 2017*; *Jaritkhuan & Suanjit, 2018*). These fatty acids are also crucial for human health and well-being (*Horrocks & Yeo, 1999*; *Osendarp, 2011*; *Asztalos et al., 2016*). Thraustochytrids can provide an alternative energy source, as through palmitic acid (PA), used in biodiesel production (*Lee Chang et al., 2013*; *Kim et al., 2013*). Production of PUFAs using thraustochytrids has been developed; thus, these organisms represent a promising and significant source of PUFAs for biotechnological applications.

For mass culturing, thraustochytrids require a high oxygen mass transfer (*Michelin et al., 2013*) to produce primary metabolites during the growth phase (*Qu et al., 2010*). Oxygen is a key substrate for aerobic bioprocesses and has low solubility in aqueous solutions. Therefore, a continuous oxygen supply is needed and the oxygen transfer rate must be known to achieve an optimum design operation in batch bioreactors (*Garcia-Ochoa & Gomez, 2009*; *Chang et al., 2013*). Bioreactors have already been developed for culturing microbial cells. Therefore, aerobic cell cultivation using bioreactors should be an effective approach to enhance mass production of thraustochytrids. However, optimizing an appropriate bioreactor will be key in achieving successful cultivation of this microorganism.

Currently, many different types of bioreactors are in use for cell culture, including stirred tank bioreactor (STB), bubble bioreactor (BB) and internal loop airlift bioreactor (ILAB). STB, the most widely-used bioreactor, is equipped with an impeller for homogenizing culture media and a sparger for delivering oxygen to the cells. STBs are primarily used to scale-up the culture process from a research and development scale to a manufacturing scale. BBs are basically cylindrical bioreactors, with a gas distributor at the bottom; whereby the gas is sparged, in the form of bubbles into either a liquid phase or a liquid-solid suspension. The BB method is normally used for cell growth and oxygen supply; because it doesn't use a complex mixing system, these reactors can be compact and have low operating and maintenance costs (*Kantarci, Borak & Ulgen, 2005*). ILAB is one type of bubble reactor, having an internal draft-tube that promotes gas–liquid mass transfer and mixing (*Saravanan, Pakshirajan & Saha, 2008*). Today's bioreactors for mass culture of microorganisms are still expensive, even without automated production control systems. In addition, knowledge is limited on thraustochytrid culture in these bioreactor types.

Growth of *A. limacinum* is limited mostly by oxygen availability in the reactor. Increasing oxygen mass transfer in bioreactors can enhance growth and fatty acids production of *A. limacinum* to levels equivalent to a shaking flask (commonly used a reference). This research was conducted in order to develop and modify a simple, low-cost bioreactor that is reliable and easy to operate for mass production of the thraustochytrid *A. limacinum* BUCHAXM 122. Three kinds of bioreactors, stirred tank bioreactor (STB), bubble bioreactor (BB) and internal loop airlift bioreactor (ILAB) were modified for mass production of *A. limacinum* BUCHAXM 122, and maximizing biomass and DHA production. The objectives of this research were to evaluate the influence of various spargers, mixing rates and aeration rates on oxygen mass transfer coefficient ($k_La$), and

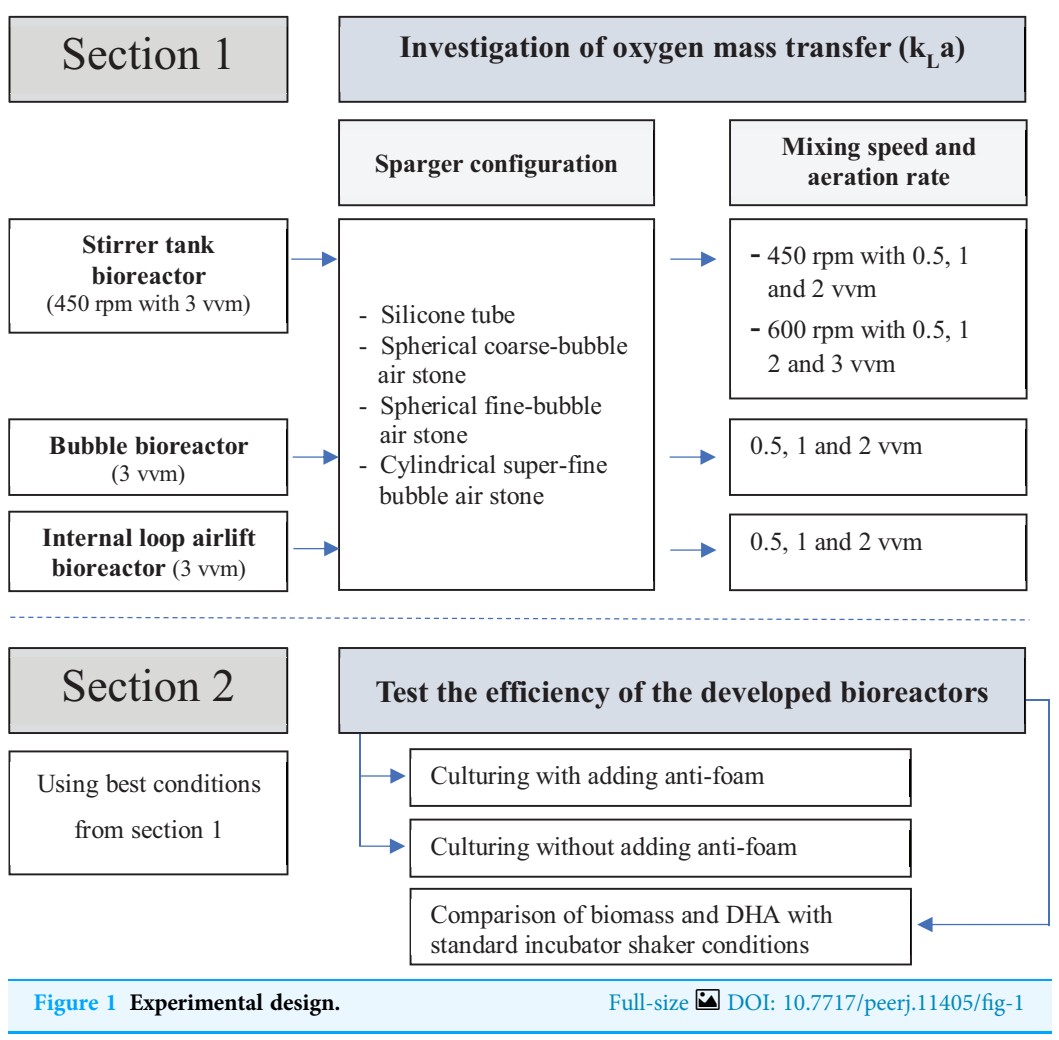

**Figure 1 Experimental design.**

then to compare biomass and DHA production in STB, BB, ILAB and incubator shaker in batch culture experiments. It should be of great interest for the biopharmaceutical industry to further continue and develop more efficient cell cultivation processes with higher product quality at lower manufacturing costs and simple operating conditions.

# MATERIALS AND METHODS

## Experimental design

Three bioreactor types, stirred tank bioreactor (STB), bubble bioreactor (BB) and internal loop airlift bioreactor (ILAB) were built and modified with the concept of low-cost and simple operation. These improved bioreactors were then evaluated for their influence on oxygen mass transfer coefficient ($k_La$), using various spargers, mixing rates and aeration rates, then compared with an incubator shaker (Section 1, evaluated using distilled water without microorganisms). Best conditions for cultivation of *A. limacinum* BUCHAXM 122 found in Section 1 (maximum $k_La$) for each developed bioreactor were determined, and compared with standard incubator shaker (Section 2). A schematic overview of this study is summarized as Fig. 1.
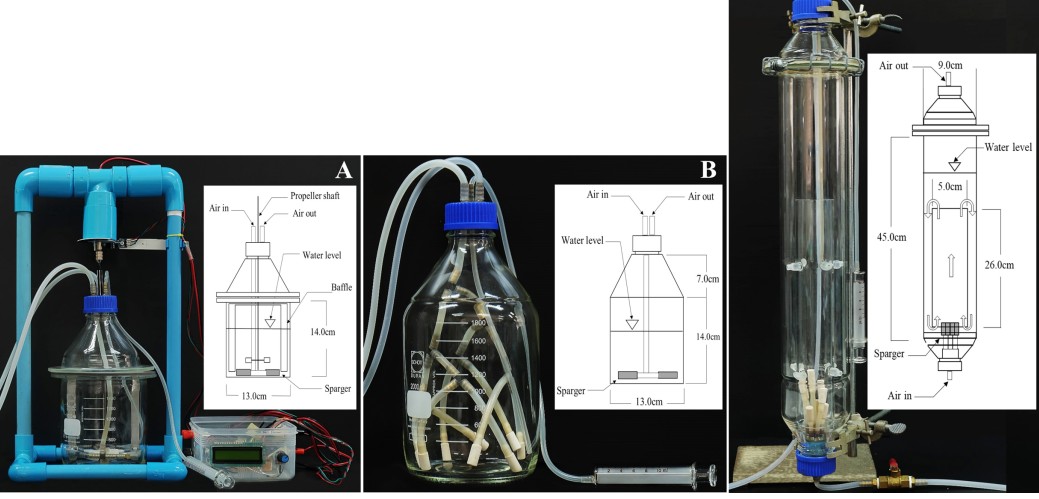

**Figure 2 Photographs and configurations of three bioreactors with spargers.** (A) Stirred tank bioreactor (STB), (B) Bubble bioreactor (BB), (C) Internal loop airlift bioreactor (ILAB).

## Bioreactor configuration and operating conditions

In this study, three types of bioreactors were designed and operated using low cost materials: stirred tank, bubble and internal loop airlift (Fig. 2). The stirred tank bioreactor (STB) was modified from *Wegrich & Shurter (1953)*, *Scragg (1991)*, and *Stanbury, Whitaker & Hall (1995)*. Our STB consisted of a culture vessel made from a 2-L laboratory bottle (working volume of 1 L), four stainless steel baffles (each baffle 1.25 cm wide × 12.5 cm high, located 0.5 cm away from the tank wall) and one set of disk turbines (5 cm in diameter, with eight 0.9 × 1 cm blades). The impeller shaft was made of a 30-cm carbon rod. Impeller rotation was controlled by a microcontroller system (Arduino). Motor speed was regulated through a Proportional-Integral-Derivative Controller (PID Controller). The aeration system consisted of a sparger installed under the impeller. Air inlet-outlet and sampling tubes were installed at the top of the bioreactor (Fig. 2A).

The bubble bioreactor (BB) was constructed using a 2-L laboratory bottle with a working volume of 1 L. The culture vessel was 13 cm in diameter and 21 cm in height. Its aeration system consisted of a sparger bundle located at the bottom of the reactor. The sparger provided uniform aeration throughout the vessel. Air inlet, outlet and sampling tubes were installed, similar to the STB (Fig. 2B).

The internal loop airlift bioreactor (ILAB) was redesigned with modification from *Cerri & Badino (2010)*. The culture vessel was 9 cm in diameter and 45 cm in height, with a working volume of 2 L. The internal draft tube was made of borosilicate glass with 26 cm height and 5 cm diameter. Aeration was provided through a sparger bundle at the bottom. Air was bubbled from the bottom through the draft tube (riser), creating water flow downward through the downcomer channel, between the draft tube and outer wall (Fig. 2C).
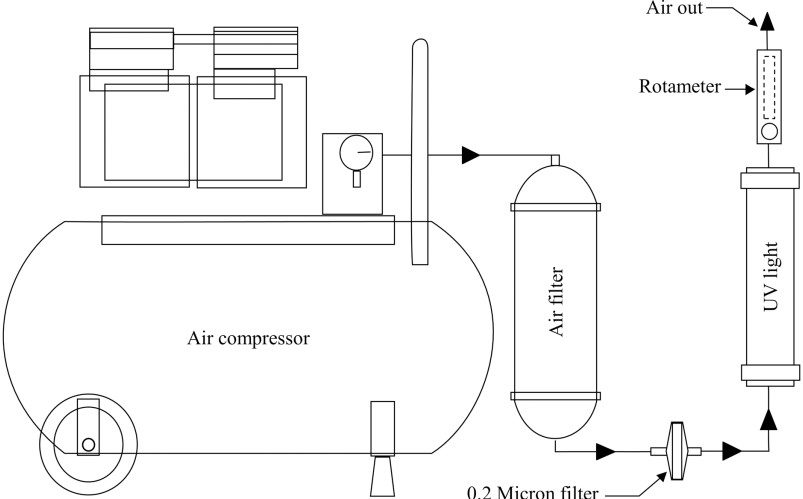

**Figure 3 Bioreactor aeration system.** Air out from the air compressor passes through two filters and UV light faces all bioreactors; aeration rate is controlled by the rotameter.

Air purification was performed using a series of two filters (0.3 μm diameter polyethylene filter and a 0.2 μm diameter filter) and 12-watt UV lamp. The filters and UV lamp were connected by a silicone tube. Prior to operation, the tube outlet end from each bioreactor was immersed in 1,000 ppm copper (II) sulfate pentahydrate ($CuSO_4.5H_2O$) to prevent outside contamination (*Reyes-Jara et al., 2016*) (Fig. 3). Aeration rate was controlled by rotameter (Dwyer, Michigan City, Indiana, USA).

## Investigation of oxygen mass transfer
### Effect of sparger configuration on oxygen mass transfer efficiency

Six sparger configurations for aeration were investigated for oxygen mass transfer capability in the bioreactors: (1) silicone tube with 4 mm internal diameter, without sparger (control), (2) spherical coarse-bubble air stone, (3) spherical fine-bubble air stone, (4) four spherical fine-bubble air stones, (5) four cylindrical super-fine bubble air stones and (6) eight cylindrical super-fine bubble air stones (Fig. 4). The coarse and fine air stones were 25 mm in diameter with pore size of 200–500 μm and 150–300 μm, respectively, while the cylindrical air stones were 10 mm in diameter and 20 mm in length, with a pore size of 10–20 μm. These air stones are widely used in aquariums. Oxygen mass transfer capability by spargers was indicated by volumetric mass transfer coefficient ($k_L a$). $k_L a$ value was measured in all bioreactors using distilled water, and then compared to the value obtained for an incubator shaker (GALLENKAMP, UK), using 250-mL Erlenmeyer flasks agitated at 200 rpm. Testing was performed using five trials. $k_L a$ was measured using a DO meter (YSI 550A, steady-state Clark type polarographic, USA) at room temperature, with zero oxygen water generated by the sulfite method (modified from *Puskeiler & Weuster-Botz, 2005*; *Bouaifi et al., 2001*). Atmospheric air containing approximately 21 percent oxygen was then introduced into all bioreactors and shakers. All dissolved oxygen measurements were continuously measured from zero oxygen

**Figure 4 Sparger types.** (A) Silicone tube, (B) Spherical coarse-bubble air stone, (C) Spherical fine-bubble air stone, (D) Cylindrical super-fine bubble air stone.

concentration until they reached saturation (average $7.04 \pm 0.16$ ppm). Data collection was performed by video recording of the DO meter display. During measurement of dissolved oxygen in the shaker, the flask was locked to the shaking platform and the oxygen probe was securely fixed with cotton wrapped in foil and masking tape at the mouth of the flask, with the tip not touching the flask bottom. In STB and BB reactors, the oxygen probe was located at the bottom of the tank, with the tip not touching the bottom. For the ILAB, the oxygen probe was affixed to the middle of the riser.

The $k_L a$ value was measured by employing the dynamic method (non-fermentative method) (*Garcia-Ochoa & Gomez, 2009*). The relationships of $k_L a$ to the oxygen transfer rate (OTR) and the oxygen uptake rate (OUR) of a growing cell were described by the following equations, (Eqs. (1)–(3)). These relationships represent one of the foundations of bioreactor design.

$$\frac{dC}{dt} = OTR - OUR \tag{1}$$

$$OTR = k_L a(C_s - C) \tag{2}$$

Substituting Eq. (2) into Eq. (1) results in

$$\frac{dC}{dt} = k_L a(C_s - C) - OUR \tag{3}$$

where $C_s$ is the saturated oxygen concentration in the liquid phase, and C is the DO concentration, OUR=0. Integration of Eq. (3) from $C_0$ to C gives:

$$\ln(C_s - C)/(C_s - C_0) = -k_L a \cdot t \tag{4}$$

where $C_0$ is the initial DO concentration of dissolved oxygen at t = 0,

The $k_L a$ value was calculated from slope by plotting $\ln (C_s - C)/(C_s - C_0)$ against time according to the equation. The $k_L a$ obtained was calculated from the linear slope of oxygen increase during aeration. These measurements were performed for around 30–90 seconds for all treatments. Calculation of $k_L a$ was based on the principle that R-squared is closest to 1.

In STB, all sparger types were tested, using an agitation speed of 450 rpm and aeration rate of 3 vvm (*Chang et al., 2014*). In BB and ILAB, all sparger configurations were tested with 3 vvm aeration (*Özcan, Sargın & Göksungur, 2014*). The bundle of four

fine-bubble air stones was not investigated in ILAB, because the space under the draft tube was too small.

## Effects of agitation speed and aeration rate on oxygen mass transfer efficiency

The experiment was performed using sparger types that provided highest $k_L a$ from the previous experiment. Bioreactor aeration was further evaluated with different agitation speeds and aeration rates. Agitation speeds tested in STB were 450 and 600 rpm (*Chang et al., 2013*; *Chang et al., 2014*), in combination with aeration rates of 0.5, 1, 2 and 3 vvm, BB and ILAB were operated at identical aeration rates.

## Cultivation of *A. limacinum* BUCHAXM 122 in bioreactors

### Stock culture and inoculation of A. limacinum BUCHAXM 122

*A. limacinum* BUCHAXM 122 was isolated from mangrove forests located in the Mangrove Resources Development Station 2, Tha Son, Chanthaburi Province, Thailand. *A. limacinum* BUCHAXM 122 inoculum was prepared in 250 mL Erlenmeyer flasks containing 50 mL of GYP liquid medium, composed of 60 g/L glucose, 10 g/L yeast extract and 10 g/L peptone in 15 PSU seawater. Cultures were incubated at 25 °C with shaking (200 rpm) for 48 h, before being used as the inoculum.

### Comparison of bioreactor types for A. limacinum BUCHAXM 122 cultivation

Five percent (5%, v/v) *A. limacinum* BUCHAXM 122 inoculum was transferred and cultured in each bioreactor, using GYP medium with the optimum sparger type, aeration rate and agitation speed obtained from previously described experiments. Shaking flask cultures were carried out in 250 mL Erlenmeyer flasks containing 50 mL of GYP medium with 200 rpm agitation, then assigned as the control. All experiments were performed in duplicate. Cell samples were collected every 24 h throughout the 168-h cultivation period, to determine biomass and DHA production. Dissolved oxygen was measured by DO meter (YSI 550A). Reduced sugar was determined by the dinitrosalicylic acid (DNS) method (*Miller, 1959*).

Dry biomass was determined by sampling 2 mL culture media from each bioreactor or culture flask, using sterile procedures. Samples were filtered through Whatman GF/C (1.2 μm) and rinsed with PBS solution. Filtered cells were then freeze dried at −80 °C (using freeze-dryer Heto LyoLab 3000, Heto-Holten A/S, Allerød, Denmark), and weighed. Average dry cell weight was plotted on a growth curve. Direct transmethylation (modified from *Shimizu et al., 1988*) was performed in order to analyze fatty acid composition of *A. limacinum* BUCHAXM 122. Extracted fatty acid methyl esters (FAME) were analyzed using gas chromatograph (HP 6890 Series GC System, Wilmington, DE, USA) equipped with a flame ionization detector (FID) and fitted with a column using capillary column HP-INNO Wax polyethylene glycol (60 m × 250 μm × 0.25 μm). Initial temperature was 50 °C for 1 min, increasing at a rate of 50 °C/min to 200 °C, holding for 1 min, then increasing by 2 °C/min to 210 °C, and holding for 14 min. Helium was used as the carrier gas. Both detector and injector temperatures were set at 250 °C. Fatty acid content was

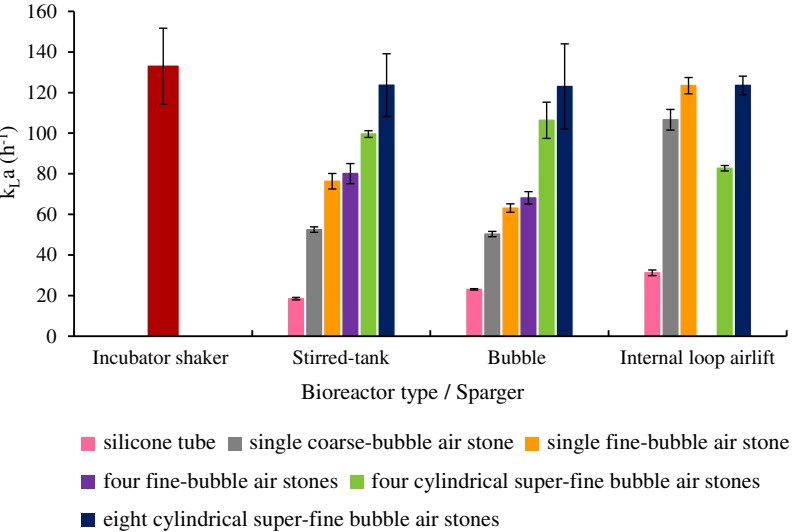

**Figure 5 $k_La$ values in three bioreactors using different sparger configurations.** All sparger types were tested, using aeration rate of 3 vvm in bubble, internal loop airlift and stirred tank bioreactor, with an agitation speed of 450 rpm.

calculated according to peak area of the chromatogram, in accordance with the internal standard (nonadecanoic acid, 19:0, Fluka, Buchs, Switzerland).

## Statistical analysis

Data were evaluated using one-way analysis of variance (ANOVA) and LSD for post hoc comparisons, at a confidence level of 5% ($p < 0.05$). Multiple linear regression (MLR) was performed with the stepwise method in order to show relationships between $k_La$ and specific growth rate, productivity, and sugar consumption.

## RESULTS

### Effect of sparger configuration on oxygen mass transfer capability

As shown in Fig. 5, aeration through a bundle of eight super-fine air stones resulted in the highest $k_La$ value for STB ($123.66 \pm 15.47$ $h^{-1}$), BB ($122.99 \pm 21.04$ $h^{-1}$) and ILAB ($123.53 \pm 4.57$ $h^{-1}$). $k_La$ values for eight super-fine air stones in STB and BB were significantly higher than for other spargers. $k_La$ values were similar for all air stone configurations in ILAB. Using optimized conditions, the $k_La$ values obtained from eight super-fine air stones were not significantly different ($p > 0.05$) from Erlenmeyer flasks in the rotary shaker at 200 rpm ($132.98 \pm 18.75$ $h^{-1}$) (Table S1).

### Effects of agitation speed and aeration rate on oxygen mass transfer

$k_La$ values in all bioreactors increased with increasing aeration rate (Table 1). An aeration rate of 3 vvm in BB and ILAB resulted in significantly higher $k_La$ values than with other rates ($p < 0.05$). In STB, the highest $k_La$ value of $155.92 \pm 18.52$ $h^{-1}$ ($p < 0.05$) was found at 600 rpm with 3 vvm; however, there were no significant differences in $k_La$ values using aeration rates of 2 and 3 vvm or agitation speeds of 450 and 600 rpm ($p > 0.05$).

**Table 1 $k_La$ values for bioreactors at different agitation speeds and aeration rates.**

| Aeration rate (vvm) | $k_La$ ($h^{-1}$) | | | |
|---|---|---|---|---|
| | Stirred | | Bubble | Internal loop airlift |
| | 450 rpm | 600 rpm | | |
| 0.5 | $51.03 \pm 2.72^a$ | $77.18 \pm 5.10^b$ | $37.70 \pm 1.45^a$ | $41.68 \pm 1.32^a$ |
| 1.0 | $82.38 \pm 3.38^b$ | $95.57 \pm 4.86^c$ | $53.96 \pm 1.13^b$ | $42.33 \pm 2.16^a$ |
| 2.0 | $129.71 \pm 5.03^d$ | $126.21 \pm 7.65^d$ | $86.50 \pm 2.93^c$ | $89.87 \pm 3.58^b$ |
| 3.0 | $123.66 \pm 15.47^d$ | $155.92 \pm 18.52^e$ | $122.99 \pm 21.04^d$ | $123.53 \pm 4.57^c$ |

Note:
Mean for each bioreactor type with different superscript letters are significantly different at $p < 0.05$.

## Growth and biomass production of *A. limacinum* BUCHAXM 122 in bioreactors

When all bioreactors culturing *A. limacinum* BUCHAXM 122 were aerated at a high rate, i.e., 3 vvm, there was a problem with foam formation on top of the culture medium. Therefore, the experiment was repeated with reduced aeration rates of 2 vvm for STB and BB and 1.5 vvm for ILAB in combination with 0.49% polypropylene glycol (SIGMA–ALDRICH) antifoaming agent. Biomass in STB peaked within 48 h ($43.05 \pm 0.35$ g/L); thereafter, the biomass decreased and remained stable from 96–120 h ($30.70 \pm 1.84$–$30.65 \pm 1.20$ g/L), with $2.07 \pm 0.12$ g/L reducing sugar residue. For ILAB, biomass rapidly increased within 48 h ($35.50 \pm 1.84$ g/L), after which it slowly and continuously increased until 120 h ($37.60 \pm 3.82$ g/L), with $3.61 \pm 2.67$ g/L reducing sugar residue. For the shaker, biomass rapidly increased within 48 h ($29.55 \pm 1.77$ g/L); after that the biomass slowly increased between 72–120 h ($31.30 \pm 0.28$–$32.90 \pm 0.28$ g/L), with $2.04 \pm 0.46$ g/L sugar residue. BB yielded the lowest biomass, rapidly increasing at 24–48 h, then slowly and continuously increasing until 120 h ($38.55 \pm 4.60$ g/L), with $41.87 \pm 0.10$ g/L reducing sugar residue. Highest biomass and residual glucose for each bioreactor are shown in Fig. 6. Comparing all bioreactors at 48 h, the highest biomass was for STB, which had significantly higher yields ($p < 0.05$) than ILAB, shaker, and BB. Sugar was nearly depleted at the end of the experiment in STB, ILAB and the shaker, but not for BB (Table S2).

Highest DHA production was obtained from ILAB at 120 h ($199.02 \pm 0.41$ mg/g DW, $35.36 \pm 2.51$% TFA). However, there was no significant difference in DHA production when comparing this treatment to ILAB at 72 h ($189.95 \pm 9.98$ mg/g DW, $36.93 \pm 0.64$% TFA), ILAB at 96 h ($189.77 \pm 7.65$ mg/g DW, $38.15 \pm 0.70$% TFA), and the shaker at 96 h ($190.50 \pm 10.64$ mg/g DW, $35.15 \pm 2.31$% TFA) ($p > 0.05$) (Fig. 7, Table S3). Initial DO in the culture medium was $0.03 \pm 0.01$ mg/L. DO values at the end of fermentation in STB ($8.20 \pm 0.09$ mg/L) and ILAB ($8.23 \pm 0.36$ mg/L) were significantly higher than in the shaker ($6.59 \pm 0.08$ mg/L) and BB ($6.95 \pm 0.16$ mg/L).

A further experiment was conducted, reducing aeration rates to 0.1 vvm for all bioreactors and the shaker, and without addition of anti-foaming agent. Highest biomass was observed in STB ($25.00 \pm 0.99$ g/L), showing no significant difference from the

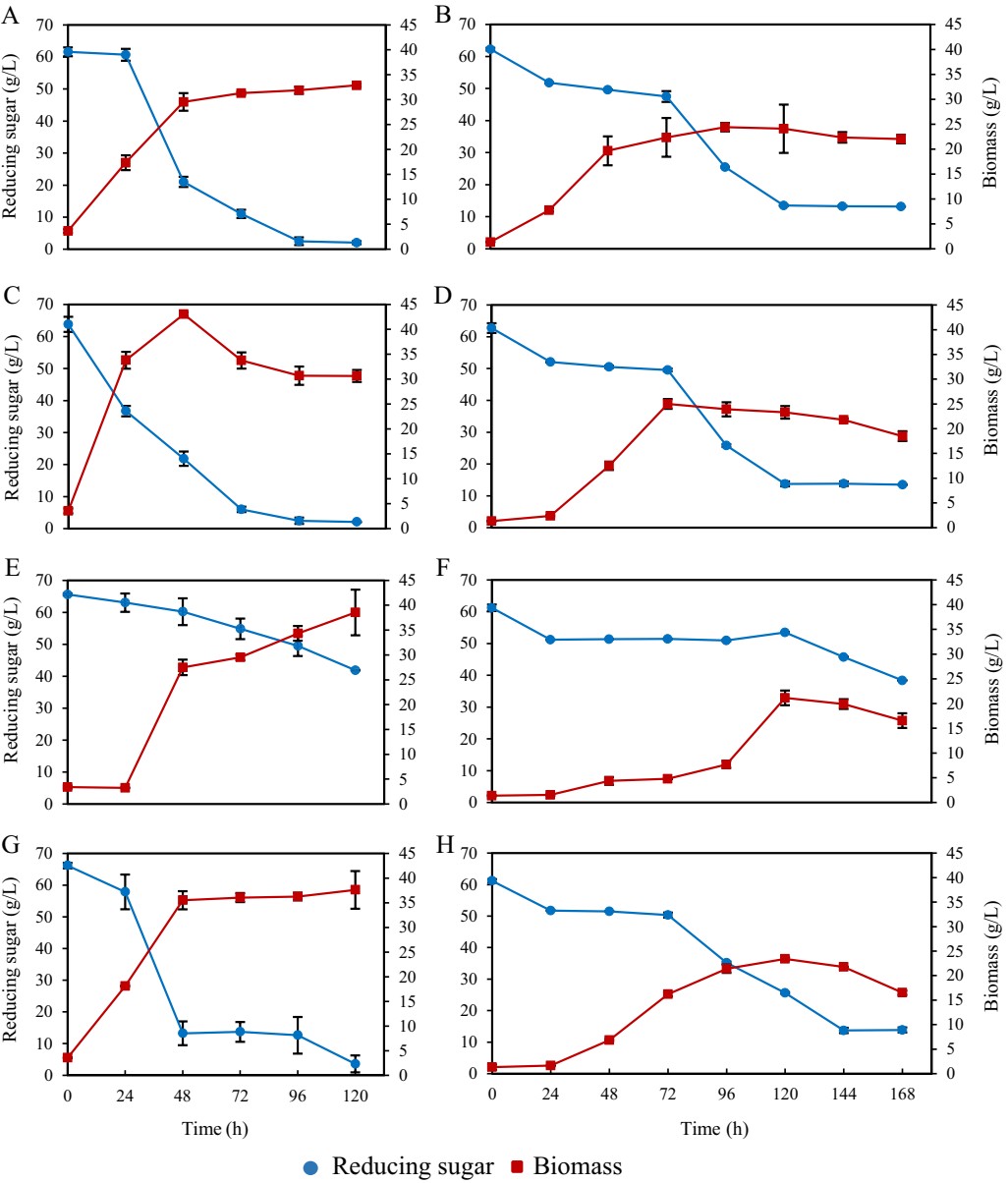

**Figure 6 Biomass and total sugar during 120 h and 168 h fermentation of *A. limacinum* BUCHAXM 122 in different bioreactors with constant initial sugar concentration (60 g/L).** (A, B) Shaker at 200 rpm; (C, D) Stirred tank bioreactor at 2 vvm and 0.1 vvm; (E, F) Bubble bioreactor at 2 vvm and 0.1 vvm; (G, H) Internal loop airlift bioreactor at 1.5 vvm and 0.1 vvm.

shaker (24.40 ± 0.85 g/L) or ILAB (23.41 ± 0.29 g/L) ($p > 0.05$). Biomass in STB was significantly higher than BB (21.15 ± 1.48 g/L) ($p < 0.05$). Moreover, there were no significant differences in DHA production among the shaker, STB and ILAB ($p > 0.05$) (Table 2). STB, shaker and ILAB consumed more glucose than BB. However, at the end of the experiment, glucose was not completely consumed in any of the treatments (Fig. 6, Table S4).

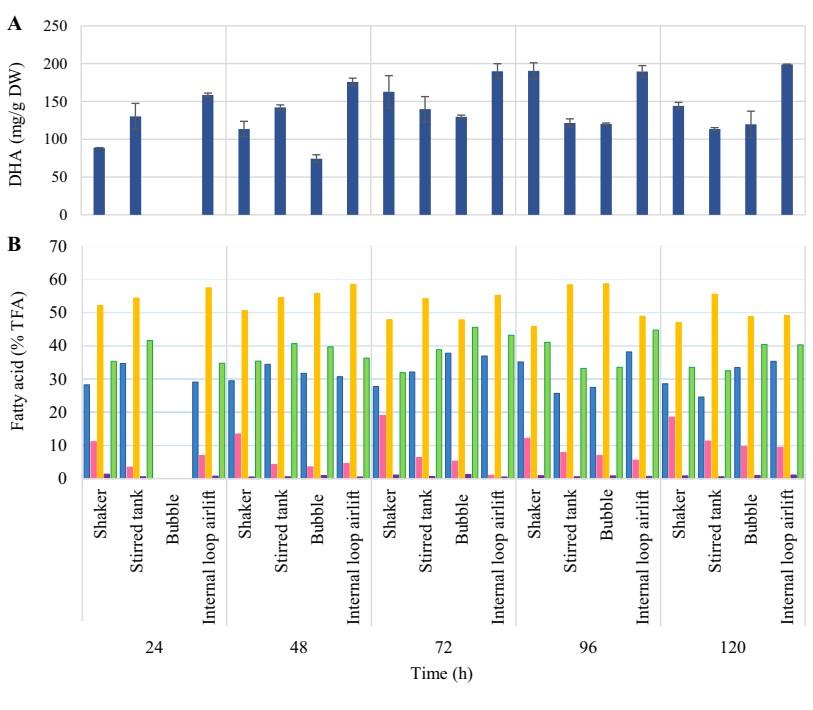

**Figure 7 DHA (mg/g DW) (A) and fatty acid (% TFA) (B) production by *A. limacinum* BUCHAXM 122 during 120 h in different bioreactor types.** Aeration provided at 2 vvm for bubble and stirred tank bioreactors with agitation speed at 600 rpm and 1.5 vvm for internal loop airlift bioreactor.

**Table 2 DHA production at 168 h in different bioreactor types, provided with aeration at 0.1 vvm.**

| Fatty acid | | Shaker | Stirred tank | Bubble | Internal loop airlift |
|---|---|---|---|---|---|
| 22:6 (DHA) | mg/g DW | 237.78 ± 35.04[a] | 217.08 ± 6.85[a] | 75.53 ± 15.79[b] | 250.48 ± 18.51[a] |
| | % TFA | 39.05 ± 3.44[a] | 34.04 ± 0.79[a] | 19.84 ± 2.86[b] | 40.20 ± 6.09[a] |
| Others | % TFA | 2.34 ± 0.10[a] | 4.38 ± 4.12[a] | 2.89 ± 0.22[a] | 11.08 ± 13.87[a] |
| SUM SFA | % TFA | 49.85 ± 3.61[a] | 52.64 ± 4.58[ab] | 68.69 ± 0.83[b] | 42. 82 ± 10.47[a] |
| SUM MUFA | % TFA | 1.13 ± 0.05[a] | 1.25 ± 0.01[a] | 3.57 ± 4.38[a] | 1.57 ± 0.37[a] |
| SUM PUFA | % TFA | 46.68 ± 3.66[a] | 41.72 ± 0.47[a] | 24.86 ± 3.33[b] | 44.52 ± 3.03[a] |

**Note:**
Means with different superscript letters (in rows) are significantly different ($p < 0.05$); FA, saturated fatty acids, MUFA, monounsaturated fatty acids, PUFA, polyunsaturated fatty acids.

DO values before and at the end of fermentation in the shaker (6.55 ± 0.03 mg/L, 6.54 ± 0.01 mg/L), STB (6.48 ± 0.02 mg/L, 6.45 ± 0.06 mg/L) and ILAB (6.45 ± 0.05 mg/L, 5.95 ± 0.21 mg/L) were quite similar, while DO for BB (4.99 ± 0.05 mg/L, 1.03 ± 0.11 mg/L), was significantly lower.

Comparison between $k_L a$ and aeration rate revealed that aeration rates in STB at 2 vvm and in BB and ILAB at 3 vvm resulted in $k_L a$ values equivalent to that in the shaker (Fig. 8). Multiple regression analysis results showed that specific growth rates of *A. limacinum* BUCHAXM 122 was positively correlated with the $k_L a$ value.

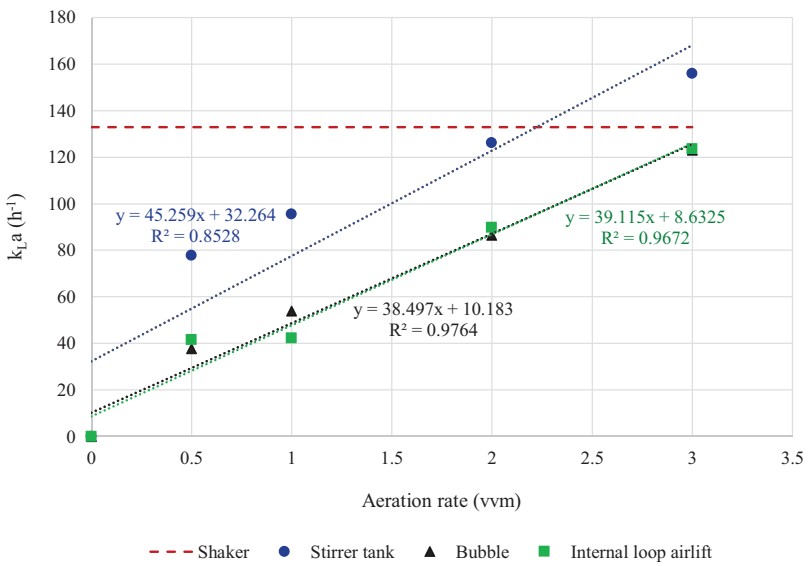

**Figure 8 Exponential relationship between aeration rate and $k_La$ in different bioreactors and shaker.** The $k_La$ values at 0.5, 1, 2 and 3 vvm in bubble, internal loop airlift and stirred tank bioreactors with agitation speeds of 600 rpm and 200 rpm in shaker.

## DISCUSSION

### Bioreactor configuration and operating conditions

Advantages of using STB, BB and ILAB bioreactors for microbial production have been reported in previous studies (*Stanbury, Whitaker & Hall, 1995*; *Cerri & Badino, 2010*). The noteworthy feature of STB is that the sparger is placed under the impeller, producing the highest $k_La$ value and resulting in a high biomass. The STB, although widely used in biological production, is well known to have shearing problems, which can cause cell damage (*Kim, Kang & Lee, 1997*). In this study, the BB design was modified by using 2-L laboratory glass bottles. It has low maintenance costs and reduces overall cost of oxygen delivery by 10–20 percent, relative to the STB (*Humbrid, Davis & McMillan, 2017*). However, in this study, its simplified design caused sedimentation in the culture vessel, resulting in the lowest biomass production. This was due to the combined effects of the culture media flow properties and the gas content, which relates back to the diameter of the reactor (*Kantarci, Borak & Ulgen, 2005*).

Results of this study showed that ILAB had a lower $k_La$ value than STB; on the other hand, liquid circulation and mixing was better than in BB. ILAB is generally used in applications involving low viscosity fluids, not requiring mechanical agitation, and providing low shear forces (*Fontana, Polidoro & da Silveira, 2009*). This study suggests that ILAB might suitably support high fat microbial culture. For instance, the ILAB bioreactor has been used to cultivate oleaginous yeast *Rhodotorula glutinis* (*Yen & Liu, 2014*). However, ILAB has limited capability when used in high aeration applications (*Qu et al., 2010*) due to foam formation. Reduced aeration rates and lower oxygen mass transfer might affect the oxygen availability for cell growth.

## Effect of sparger configuration on capability of oxygen mass transfer

This study shows that the bundle of eight super-fine air stones was the best sparger type for all three bioreactors, since it produced a $k_La$ value equivalent to the standard 200 rpm rotary shaker ($132.98 \pm 18.75$ $h^{-1}$). The $k_La$ value measured from rotary shaker was similarly to the $k_La$ value ($108$–$180$ $h^{-1}$) obtained by shaking in a 50-mL sodium sulfite system at 120–200 rpm (*Kato et al., 2005*). The super-fine pores provide numerous small air bubbles, with uniform distribution by using eight stones, leading to a homogeneous flow. This homogeneous flow facilitates the biochemical reactions involved in oxygen utilization and mass transfer of oxygen to cells (*Garcia-Ochoa et al., 2010*), while larger bubbles lead to coalescing and unsteady heterogeneous flow creation (*Hoseinkhani et al., 2019*). Results of this study are consistent with a previous report, which found that increasing surface area of the sparger and number of pores resulted in higher $k_La$ (*De Wilde et al., 2014*) by improving bubble dispersion, preventing bubble overflow and reducing gas flow rate. Similarly, cultivation of *Schizochytrium* sp. using a microbubble-type sparger was found to have 8.6-fold higher $k_La$ values than with a general type sparger (*Ju et al., 2019*). However, it was also based on the pore size of sparger; if the pore size is too large, the $k_La$ difference may not be seen. For instance, the $k_La$ value was not significantly different when using spargers with different numbers of holes (1, 4 and 9) and hole diameters (6, 3 and 2 mm) (*Karimi et al., 2013*). Conversely, the sparger affects bubble size, and bubble size is specific to certain oleaginous microalgae culturing. For instance, culturing *Crypthecodinium cohnii* in BB using bubbles of 0.18 cm diameter resulted in higher biomass than bubbles of 0.09 and 0.36 cm diameter. However, the 0.36 cm bubbles gave highest DHA production, followed by 0.18 and 0.09 cm, respectively (*Hoseinkhani et al., 2019*). This indicates that there is an optimal bubble size leading to growth and production of healthy fatty acids. For this reason, further studies to find the appropriate sizing of air bubbles for *A. limacinum* BUCHAXM 122 production should be initiated.

## Effects of agitation speed and aeration rate on oxygen mass transfer

This study shows that increasing agitation speed and aeration rate resulted in higher $k_La$ values. Similar results have been reported for a 30-L STB system; when agitation speed was increased from 300 rpm and 0.6 $m^3$/h to 600 rpm and 2 $m^3$/h, $k_La$ value increased from $143 \pm 19$ $h^{-1}$ to $1{,}802 \pm 105$ $h^{-1}$ (*Chang et al., 2013*). In addition, higher agitation speeds (400–800 rpm) with 1–3 L/m have been found to be the most efficient for oxygen mass transfer in STB (*Karimi et al., 2013*). Similar to our study using agitation speeds of 450 and 600 rpm, with aeration rate of 1–3 vvm. Agitation rates below 300 rpm resulted in insufficient agitation to capture air bubbles in the bioreactor, and consequently may limit mass transfer of oxygen. Our data show that aeration rate has an effect on the $k_La$ value in ILAB and BB, as well. Likewise, in the cultivation of *Rhodotorula glutinis* in ILAB, it was found that $k_La$ value increased with aeration rate (1.0, 1.5, 2.0 and 2.5 vvm), and the aeration rate at 1.5–2.5 vvm was able to provide high biomass and fatty acid yields (*Yen & Liu, 2014*). Higher $k_La$ values have also been obtained in bubble column bioreactor (BCB) by increasing the aeration rate (*Kumar & Vinod, 2014*).

Effects of agitation speed and aeration rate on $k_La$ can be explained by the relation of stirrer speed and specific interfacial area values in STB, and by bubble size in all bioreactor types. Therefore, speed of the stirrer and mixing play an important role in breaking up the air bubbles. Also, turbulence and shear are caused by baffles (*Kawase & Moo-Young, 1990*; *Garcia-Ochoa & Gomez, 2009*). Air bubble dispersion is also related to gas dispersion, gas hold-up and residence time (*Joshi, Pandit & Sharma, 1982*; *Barigou & Greaves, 1996*; *Garcia-Ochoa & Gomez, 2009*). Dispersion is also related to other factors, such as liquid properties, gas velocity, gas injection rate, operating pressure, injection tube diameter and column diameter (*Kawase, Halard & Moo-Young, 1992*; *Garcia-Ochoa & Gomez, 2009*; *Garcia-Ochoa et al., 2010*; *Mohagheghian & Elbing, 2018*).

## Biomass and DHA production of *A. limacinum* BUCHAXM 122 in bioreactors

Production of *A. limacinum* BUCHAXM 122 in STB showed faster growth and higher biomass yield than in the rotary shaker, ILAB and BB. These findings are consistent with batch fermentation of *Aspergillus oryzae* in STB, with agitation rates of 200–700 rpm and aeration rates of 0.5–2.5 vvm (14.1 ± 0.13 g/L) which is higher than ILAB at 0.5–2.5 vvm (10.1 ± 0.03 g/L) (*Fontana, Polidoro & da Silveira, 2009*). In general, increases in $k_La$ lead to more biomass production (*Hoseinkhani et al., 2019*). This is due to oxygen demand, particularly in the ATP production process. Cell density increases rapidly during the exponential phase, producing a large number of primary substances (primary metabolites), such as enzymes, nucleic acids and other proteins from ATP by aerobic respiration (*Qu et al., 2010*). Conversely, when DO concentrations decrease, metabolic rate of microorganisms displays a significant decrease (*Sinha, Dautz & Hader, 2001*). From experiments with an STB system, fed-batch cultures of *Schizochytrium* sp. S31 with agitation rates of 300, 450 and 600 rpm revealed that 600 rpm provided highest oxygen uptake rates and yielded the highest biomass (*Chang et al., 2014*). These findings are consistent with our study. Comparing two other bioreactors (*Özcan, Sargın & Göksungur, 2014*) demonstrated that *Aureobasidium pullulans* produced higher biomass from ILAB, with aeration rates at 1 vvm than from BCB at 0.2, 1.1 and 2 vvm.

According to a previous study, cell growth (μ; 0.090/h) and glucose consumption rate (qs; 0.115/h) in airlift-type bioreactors were higher than from fermentation in stirrer-type vessels (0.057/h and 0.064/h, respectively) (*Hong et al., 2013*). Our results clearly indicate that biomass production from bioreactors is related to glucose consumption. The highest biomass was in STB, which also had the highest glucose consumption. BB, on the other hand, yielded the lowest biomass, with its glucose consumption being lower than the other bioreactors and the rotary shaker. This shows that efficiency of the bioreactor affects the remaining glucose in the reactor; with glucose remaining in a reactor tank being the starting point for further development of suitable culture media recipes. In addition, cell culture techniques in the log phase and reduction of oxygen during the stationary phase of fatty acid production should be further developed (*Qu et al., 2010*; *Rosa et al., 2010*).
One of the limitations of using a high aeration rate is that it produces large amounts of foam. In our cultures of *A. limacinum* BUCHAXM 122, higher biomass was obtained at 2 vvm in STB and BB and at 1.5 vvm in ILAB. However, if an antifoaming agent is required to avoid overflowing, it can cause a decrease in $k_L a$ values (*Sundaramurthy, Srivastava & Mishra, 2011*). Consequently, cell cultivation in STB could be carried out at aeration rates of 0.1 vvm, since biomass concentrations for 72 h (25.00 ± 0.99 g/L) were similar to the shaker (22.35 ± 3.89 g/L), and were higher than the range of values reported for *A. limacinum* (range of 9.39–20.71 g/L) (*Jaritkhuan & Suanjit, 2018*).

DHA production obtained from ILAB and the rotary shaker were relatively similar in yield. Highest DHA content in ILAB with 1.5 vvm aeration rate was significantly higher than both STB and BB. It was found that high agitation speeds negatively impacted DHA production in STB. As reported by *Chang et al. (2014)* for *Schizochytrium* sp. S31, an agitation speed of 600 rpm produced lower DHA (29.37 ± 0.89% TFA) than 300 rpm (36.04 ± 1.78% TFA). Our experiments also showed that using an agitation speed of 600 rpm, STB produced lower DHA yield than other bioreactor types.

It is known that microbial cells are sensitive to mechanical shear stress, particularly from high agitation intensity, as it causes cell damage (*Wase et al., 1985*; *Hong et al., 2013*). The cell wall of thraustochytrid organisms is non-cellulosic and thin, ranging from 2 to 3 nm thick (*Marchan et al., 2018*), and can be affected by relatively high shear stress, consistent with the report by *Wang & Lan (2018)*. In addition, cells with flagella are more shear sensitive than unicellular algae cells (*Wang & Lan, 2018*). Thraustochytrids can move during the zoospore stage using two flagella. They may be affected by shear, resulting in less fatty acid production, since zoospores also store fatty acids (*Morita et al., 2006*).

A low aeration rate is more favorable for efficient DHA synthesis (*Qu et al., 2010*; *Hoseinkhani et al., 2019*). This study demonstrates that low aeration rates, i.e., 0.1 vvm, provide highest DHA yield during late stationary phase (168 h) in all bioreactors. ILAB, operated under low aeration rate (with less shear stress), may be an appropriate choice for DHA production. High oxygen content during the cell stage accumulates fatty acids, allowing cells to use carbon sources (glucose) in the breathing process and metabolism, rather than being used in lipid accumulation processes. The appropriate amount of primary metabolite production by oxygen limitation produces a sufficient quantity of Acetyl-CoA (a precursor to the synthesis of fatty acids). This also results in higher cell accumulation of fatty acids. Therefore, the DHA content in thraustochytrids depends on both amount of biomass (log and early stationary phase) and production of fatty acids (stationary phase). According to this study, both ILAB and shaker have lower shear forces than STB, which has a positive effect on maintaining cellular integrity. The cells are not damaged, resulting in a high DHA content. Moreover, the results show that $k_L a$ value in ILAB was relatively lower than in the shaker. It is also associated with synthesis of fatty acids for each strain of thraustochytrid, as a pathway of with or without oxygen, to comply with the in-cell enzyme system (*Jakobsen et al., 2008*).

Higher DHA production may be related to the activity of anaerobic fatty acid pathways (PUFA synthase or PKS), highly active under low dissolved oxygen (*Matsuda et al., 2012*; *Metz et al., 2001*). Studies on transcriptome and gene expression analysis of

**Table 3 Biomass and DHA yields for *Aurantiochytrium* sp. grown with different bioreactor types and culture media.**

| Bioreactor type | Strain | Carbon and nitrogen source | Fermentation volume (L) and conditions | Biomass (g/L) | DHA (% TFA) | Reference |
|---|---|---|---|---|---|---|
| Stirrer tank | *A. limacinum* SR21[a] | Glucose (NH$_4$)$_2$SO$_4$ Corn steep liquor | 3 L (Fed batch) 300 rpm | 21.9–59.2 (62–125 h) | 33.3–38.6 (92 h) | *Yaguchi et al. (1997)* |
| | *A. limacinum* SR21[a] | Glucose Corn steep liquor Ammonium acetate | 3.5 L 300–500 rpm, 1.71 vvm, 28 °C | 51.95 (72 h) | | *Rosa et al., 2010* |
| | *A. limacinum* SR21[a] (ATCC MYA-1381) | Crude glycerol YP | 4.5 L (Continuous) 0.1 vvm | 13 (after 72 h) | 31.09 (after 72 h) | *Ethier et al. (2011)* |
| | *Aurantiochytrium* sp. KRS101 | GY | 3,000 L | 30.5 (after 48 h) | 39.5 (after 48 h) | *Hong et al. (2013)* |
| | *Aurantiochytrium* sp. TC20 | GYP | 1.6 L (Fed batch) 300–1,150 rpm, 0.3–1.5 vvm, 20 °C | 56.6 (69 h) | 43.2 (69 h) | *Lee Chang et al. (2013)* |
| | *A. limacinum* BUCHAXM 122 | GYP | 1 L 600 rpm, 25 °C 0.1 vvm | 25.00 (72 h) | 34.04 (168 h) | This study |
| | | | 2 vvm | 43.05 (48 h) | 34.47 (48 h) | |
| Bubble | *A. limacinum* BUCHAXM 122 | GYP | 1 L 25 °C, 0.1 vvm | 21.15 (120 h) | 19.84 (168 h) | This study |
| | | | 2 vvm | 29.50 (72 h) | 37.82 (72 h) | |
| Internal loop airlift | *Aurantiochytrium* sp. KRS101 | GY | 3,000 L | 25 (after 24 h) | 52.3 (36) | *Hong et al. (2013)* |
| | *A. limacinum* BUCHAXM 122 | GYP | 2 L, 25 °C, 0.1 vvm | 23.41 (120 h) | 40.20 (168 h) | This study |
| | | | 1.5 vvm | 36.05 (72 h) | 36.93 (72 h) | |

**Note:**
[a] *Schizochytrium* (now called *Aurantiochytrium*), Numbers in parentheses following biomass and DHA yields represent fermentation time (hours).

*Schizochytrium* sp. under different oxygen supply conditions have shown that high oxygen supply is used in NADPH and acetyl-CoA production for cell growth, rather than for DHA production (*Bi et al., 2018*). Consequently, high k$_L$a is required for growth, while DHA production does not require a very high oxygen supply (*Chang et al., 2014*).

Therefore, ILAB can be used as an alternative to the STB bioreactor, with production performance similar to the standard rotary shaker. However, STB is still the most suitable bioreactor for thraustochytrids if biomass yield is the major priority, followed by production of intracellular products (*Hoseinkhani et al., 2019*). The simplified and low-cost ILAB design used in this study shows favorable production from *Aurantiochytrium* with 23.41–36.05 g/L biomass and 36.93–40.20% DHA in total fatty acids. This production yield is comparable to previous studies, which reported ranges of 13–59.2 g/L biomass (72 and 125 h) and 31.09–52.3% TFA (72 and 36 h) (Table 3) (*Yaguchi et al., 1997*; *Rosa et al., 2010*; *Ethier et al., 2011*; *Hong et al., 2013*; *Lee Chang et al., 2013*).

## CONCLUSIONS

Three designs of low-cost bioreactors, stirred tank (STB), bubble (BB) and air-lift (ILAB) were evaluated for cultivation of *Aurantiochytrium limacinum* and DHA production, in combination with optimized sparger (eight super-fine bubble air stones) and aeration rate (between 1.5–2 vvm). All reactors are capable of thraustochytrid cultivation, with growth and yield comparable to a 200-rpm rotary shaking flask. Each bioreactor type has unique advantages. STB had the highest $k_La$ values (155.92 $\pm$ 18.52 $h^{-1}$), producing the highest biomass (43.05 $\pm$ 0.35 g/L). However, agitation by the impeller causes shear stress damage to the cells, and may affect fatty acid production. BB is simple in design, easiest to build, and lowest cost. However, thraustochytrid cultivation in BB requires a high airflow rate in order to avoid cell sedimentation during cultivation. ILAB provided moderate cell density (37.60 $\pm$ 3.82 g/L) but the highest amount of fatty acids (199.02 $\pm$ 0.41 mg/g DW, 35.36 $\pm$ 2.51% TFA). With ILAB, the uniform flow pattern through the riser and downcomer in the bioreactor allows homogeneous mixing, and hence is appropriate for oleaginous cell growth. For further study, other equipment designs should be investigated that could reduce cell sedimentation, as found in BB and ILAB bioreactors. Moreover, the shape of ILAB might be redesigned, in order to accommodate the fine bubbles formed and higher air flow rate, which induces high amounts of foaming.

## ACKNOWLEDGEMENTS

The authors are grateful for the technical assistance from the staff of the Major of Environmental Science and Department of Aquatic Science, Faculty of Science, Burapha University, Chonburi, Thailand.

### Funding

This research was supported by the Faculty of Science, Burapha University. No additional external funding was received for this study. The funders had no role in study design, data collection and analysis, decision to publish, or preparation of the manuscript.

### Grant Disclosures

The following grant information was disclosed by the authors:
Burapha University.

### Competing Interests

The authors declare that they have no competing interests.

### Author Contributions

- Khanoksinee Sirirak conceived and designed the experiments, performed the experiments, analyzed the data, prepared figures and/or tables, authored or reviewed drafts of the paper, and approved the final draft.

- Sorawit Powtongsook conceived and designed the experiments, analyzed the data, prepared figures and/or tables, authored or reviewed drafts of the paper, and approved the final draft.
- Sudarat Suanjit conceived and designed the experiments, analyzed the data, prepared figures and/or tables, and approved the final draft.
- Somtawin Jaritkhuan conceived and designed the experiments, analyzed the data, prepared figures and/or tables, authored or reviewed drafts of the paper, and approved the final draft.

## Data Availability

The raw measurements are available in the Supplemental Files.

## Supplemental Information

Supplemental information for this article can be found online at http://dx.doi.org/10.7717/peerj.11405#supplemental-information.

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
