# Peer review of "Effectiveness of various bioreactors for thraustochytrid culture and production (Aurantiochytruim limacinum BUCHAXM 122)"

_PeerJ, doi:10.7717/peerj.11405_

## Round 0.1 · original submission · Major Revisions

Please take particular consideration to the comments raised about the methodology and results presentation.

·

Basic reporting

This work is a nice study on the bioreaction engineering area concerning mass transfer in bioreactors. The manuscript is clear, and well constructed. The outcomes are valuable and worthy to be published.

My main critic is that this work is not one of neither optimization nor design, but one of analysis/study about how different available aeration devices affects the performance of different bioreactors on a particular cell growth and bio-product production. So, the title does not reflect the real matter of the work.

In the hand, while the experimental aspects are well described, as well as the outcomes, there is no a related state-of-the art regarding the particular microorganism in bioreactors. Therefore, the work is likely to seam just a report of experiments. I mean, the scientific value is lost since this kind of tasks are reported even in books (e.g. Bioreaction Engineering Principles. by Villadesen, Nielsen and Lidén, Springer).

Experimental design

The experiments are well described, as well as the outcomes.

Performing diverse levels on factors is pretty expense; but an improved explanation of how these were set is convenient as well.

Validity of the findings

I think it is convenient to report the saturation concentration of oxygen, as well as the oxygen composition of injected air.

Although the increase of reactor size implies an increase on the air injection, it seems to me the resulted values of the transfer coefficient are low. Indeed a reference points how the mass transfer coeffcient increases in a stirrer tank reactor, with values greater than the ones reported here.

I found valuable the comparison of performance between reactor types.

Additional comments

The work is valuable to be published, but in its own and fair field of study. I mean, the work analyses and compare devices of aeration and bioreactors for a particular microorganism, but it does not contain neither concrete guides of design nor optimization tasks.

Reviewer 2 ·

Basic reporting

The manuscript aims to show the design of three types of low-cost bioreactors; stirred tank (STB), bubble column (BCB) and air-lift (ILAB) were evaluated for the cultivation and DHA production of Aurantiochytrium limacinum in combination with optimized sparger.
The manuscript is well written and use a clear and technically correct words.
The use of references is sufficient and relevant references are used.
The structure in general is correct and the authors might change some tables for figures, for example table 2 will be better presented as figure.
The manuscript perse do not present an hypothesis, due to the fact that is just a comparison of 4 bioreactors: Shake flasks, stirred tank bioreactor, bubble column and airlift bioreactor.
Some concern about experimentation are expressed in the next paragraph.

Experimental design

In the materials and methods the authors omit many details that should be taken into account so that future researchers can reproduce their results.
The presentation of the bioreactors is not complete, and in addition to Figure 1, the presentation of photographs of the same can be beneficial.
The Bubble Column Bioreactor is not really a bubble column and should be called something like Bubbled Bioreactor but not Column. The ratio of the fill volume height to the tank diameter does not meet the definition of a bubble column.
The authors do not present the type of electrodes they use to evaluate dissolved oxygen and evaluate KLa in shaken flasks and in the three types of bioreactor. It is vitally important to present the electrodes that were used and if those readings were made online and offline.
Did the authors always use the YSI dissolved oxygen meter that they report?
If all the dissolved oxygen measurements in flasks and in the bioreactor were done with YSI equipment, were these measurements offline?
If the YSI electrode was put in the shaken flasks, how to avoid the baffle effect of the same electrode?
How to avoid the problem that it had to be able to measure perfectly the oxygen changes in the flask?
What is the response time of the dissolved oxygen electrode ?,
Is this not affecting the Kla measurements, causing all the bioreactors to reach very close and similar KLa values ​​?.
Is not the similarity of the measured KLa an artifact of the electrode?
Why was the KLa not measured in the culture medium?, The authors of the manuscript are invited to do the Kla measurement in the culture medium and compare the differences and trends in each bioreactor when doing it only in water or with the culture medium. The relationship is not linear. The KLa measured in distilled water and in the culture medium (which is rich in nutrients and especially in a carbon source) must be completely different and also behave differently depending on the bioreactor used.

Validity of the findings

The validity of the results found in the work and the relationship of KLa with the growth of its micro-organism and the production results will depend on how the KLa measurements were made in the bioreactors. At this moment, it is difficult for this reviewer to be able to consider that the results are really related or not to mass transfer phenomena.

This section can be discussed when it is formally clarified how volumetric mass transfer coefficients were measured and calculated.
The comparison in figure 4 is not clear in the sense of why or how the experiments were done. Comparison of four culture systems under conditions. A fair comparison of the 4 culture systems should be for example at the same value of the volumetric oxygen transfer coefficient or for example at the same delivered gassed volumetric power. The authors are unclear, or at least to this reviewer why they made that decision.
The figure captions are poorly written. The authors should explain each figure caption in detail, for example figure 3 is not clear about the experimental conditions of these results.
In many of the kinetics of figure 4, standard deviations are not seen, were all the experiments done in triplicate?

Additional comments

Although the article does not present highly novel results for the biotechnology field based on the design of low-cost bioreactors, it is a well-prepared manuscript, but it omits details such as those previously commented. This particular journal does not evaluate the novelty aspects of a manuscript and therefore this reviewer has not commented on this.

Reviewer 3 ·

Basic reporting

In this article titled “Design and optimization of bioreactors based on oxygen transfer efficiency for biomass and DHA production of Aurantiochytrium limacinum BUCHAXM 122”, authors have fabricated three reactors (STR, BCB, ILAF) at low cost and chosen optimum aeration rate, sparger and agitation speed for kLa values. Then, the reactors were compared for maximum biomass and DHA production at two different aeration rates.

The following comments need to be addressed while revising the manuscript.


Basic Reporting

Author should look deep into the manuscript for grammatical errors and rectify it

Literature is well referenced in introduction and discussion

Structure is clear and conforms to PeerJ standards

In Abstract, line 4, Author has mentioned, “The bioreactors were evaluated for their influence on oxygen mass transfer coefficient (kLa) using various spargers, agitation, and aeration”. In case of BLB and ILAB, there will be no agitators which makes the statement ambiguous. Also consider modifying aeration with aeration rate.

In Abstract, line 12, author can consider the rephrasing “especially the reactors with air bubbles”.

Experimental design

Experimental Design

In materials and methods, the materials used for sparging could have been analysed for pore size through SEM for better understanding.

Line 108: The equation for determining kLa is missing closed parenthesis. Please give upper and lower limit for solving the equation.

As indicated in manuscript, DHA production is in late stationary phase. Hence author could have reduced oxygen level once it reaches stationary phase to have high productivity.

Authors should develop empirical model correlating Kla vs other operating paraments and need to validate.

Table 2 is cumbersome and difficult to interpret the data. Authors need to improve the presentation of data

Table 3 and line # 198, What is the rationale behind conducting the experiment with 0.1 vvm and in table data for all other time points are not provided. Authors should provide all the data

Power input calculations is missing. It is important to compare which process is economical.

Validity of the findings

In figure 3, the graph shows there is no significant difference between the Eight bundle super-fine air stone between control and reactors. But in line 258, author has mentioned agitator speed has positive effect on kLa values. How the authors validate the contradiction?

In Figure 4, A and B refer to shake flask experiment at 200 RPM. But Fig 4A shows has less residual glucose and more biomass than Fig 4B at 120 h. Can you elaborate on this figure?

During DHA production, please discuss on the incomplete consumption of glucose at the end of treatment?

Internal loop airlift bioreactor produces ~40 % DHA which is equivalent to shake flask yield. What is the key factor that makes IALF better than shake flask?

Additional comments

Line 68: The reactor models for STB, BCB, ILAB exists already. So, consider changing the contextual meaning of the statement.

Please mention the basis of selecting these three bioreactors in introduction.

Please mention the pore size of silicon tube sparger (control)

Line 130: PSU should be in capital letters

Line 149: Please mention the column specification and running condition for Gas chromatography. Please give reference if applicable.

In Line 257, the text is missing with the word “agitation speed” but has mentioned 300 and 600 rpm. Please include it.

Mention the aeration rate in figure 3 caption.

Significance level indicated in table with alphabets in superscript is not clear. Author should define each alphabet with their significance level.

In Table 4, Why there is difference in fermentation time for biomass concentration and DHA concentration. If it specifies individual maximum, please indicate in the referred text location.

---

## Round 0.2 · accepted · Accept

Both reviewers indicated that the manuscript was significantly improved and is now suitable for publication in PeerJ.

·

Basic reporting

In my review, my main comment was about the congruence between the title and the matter of the work, and that the manuscript presented valuable results on the field of bioreactor engineering. Also I release some additional comments to clarify some points.

In this corrected version, the title was changed in order to make it congruent with the work. I agree with the new title.
In other hand, I see that particular comments were clarified and corresponding information was included in the manuscript.

Then, I consider the manuscript as valuable to be published.

Experimental design

In this corrected version, there were included more explanations about the experimental design. I consider all of the as convenient, in such a way the purposes and procedures of experimentation can be followed.

Validity of the findings

In this corrected version, explanation of results are better backed with references, in such a way the conclusions are well stated and linked to results.

Additional comments

The manuscript is susceptible of improvement. I am not an expert in English language, so I do not dare to point sentences and paragraphs, but I felt that some lines could be better written.

Reviewer 2 ·

Basic reporting

The authors did their best to respond to the comments made by this reviewer.

Experimental design

The authors did their best to respond to the comments made by this reviewer.

Validity of the findings

The authors did their best to respond to the comments made by this reviewer.

Additional comments

The authors did their best to respond to the comments made by this reviewer.